# Resveratrol against Cervical Cancer: Evidence from In Vitro and In Vivo Studies

**DOI:** 10.3390/nu14245273

**Published:** 2022-12-10

**Authors:** Matteo Nadile, Maria Ilektra Retsidou, Katerina Gioti, Apostolos Beloukas, Evangelia Tsiani

**Affiliations:** 1Department of Health Sciences, Faculty of Applied Health Sciences, Brock University, St. Catharines, ON L2S 3A1, Canada; 2Department of Biomedical Sciences, School of Health Sciences, University of West Attica, 12243 Athens, Greece; 3National AIDS Reference Centre of Southern Greece, School of Public Health, University of West Attica, 11521 Athens, Greece; 4Centre for Bone and Muscle Health, Applied Health Sciences, Brock University, St. Catharines, ON L2S 3A1, Canada

**Keywords:** cervical cancer, resveratrol, proliferation, survival, apoptosis, signaling cascades

## Abstract

Cervical cancer affects many women worldwide, with more than 500,000 cases diagnosed and approximately 300,000 deaths each year. Resveratrol is a natural substance of the class of phytoalexins with a basic structure of stilbenes and has recently drawn scientific attention due to its anticancer properties. The purpose of this review is to examine the effectiveness of resveratrol against cervical cancer. All available in vitro and in vivo studies on cervical cancer were critically reviewed. Many studies utilizing cervical cancer cells in culture reported a reduction in proliferation, cell cycle arrest, and induction of apoptosis. Apart from apoptosis, induction of autophagy was seen in some studies. Importantly, many studies have shown a reduction in the HPV oncoproteins E6 and E7 and increased levels of the tumor suppressor p53 with resveratrol treatment. A few studies examined the effects of resveratrol administration in mice ectopic-xenografted with cervical cancer cells showing reduced tumor volume and weight. Overall, the scientific data show that resveratrol has the ability to target/inhibit certain signaling molecules (EGFR, VEGFR, PKC, JNK, ERK, NF-kB, and STAT3) involved in cervical cancer cell proliferation and survival. Further in vivo experiments and clinical studies are required to better understand the potential of resveratrol against cervical cancer.

## 1. Introduction

### 1.1. Cervical Cancer

Cervical cancer, according to data from the World Health Organization (WHO), is the fourth most common cancer in women. Worldwide, an estimated 570,000 women were diagnosed with cervical cancer, and about 311,000 women died from the disease in 2018 [1]. Cervical cancer develops in the cervix of the uterus, and almost all (99%) cases are associated with human papillomaviruses (HPV) infection [1]. Squamous cell carcinoma and adenocarcinoma are the main classifications of cervical cancer, with most of the cases seen in clinics, representing squamous cell carcinomas [2].

Cancer cell survival, proliferation, and metastasis are driven by excessive growth factor receptor signaling [3]. For example, excessive epidermal growth factor receptor (EGFR) activation due to mutations leads to overactivation of the phosphatidylinositol 3-kinase (PI3K)/protein kinase B (Akt) and Ras-mitogen activated protein kinase (MAPK) signaling pathways that lead to proliferation and survival [4]. In addition, the over-activation of these signaling cascades contributes to chemo- and radio-therapy resistance [5,6]. c-Jun *N*-terminal kinases (JNK) and extracellular signal-regulated kinase (ERK) belong to the MAPK family and have been found to be overactivated in cervical cancer [7,8]. Apart from the above, signaling molecules such as signal transducers and activators of transcription (STAT 3) [9], protein kinase C (PKC) [10], and the nuclear factor kappa-light-chain-enhancer of activated B cells (NF-kB) [11] play a role in cervical carcinogenesis. The HPV oncoproteins 5, 6, and 7 are found in many cervical cancers, and evidence indicates that they activate signaling cascades leading to proliferation, survival, and metastasis [12]. Apart from oncogenes, mutations in tumor suppressor genes such as p53 result in loss-of-function leading to inhibition of apoptosis, resulting in survival and carcinogenesis [13].

Treatment of cervical cancer involves surgery or radiation combined with chemotherapy for the earliest stages of the disease. For more advanced disease/later stages, radiation combined with chemotherapy is usually the main treatment strategy. Chemotherapy, as the stand-alone approach, is often used to treat advanced cervical cancer. Chemotherapy drugs used for cervical cancer include paclitaxel, carboplatinum, cisplatinum (CDDP), bleomycin, mitomycin-C, vincristine, and irinotecan [14,15,16]. The five-year survival rate of patients with cervical cancer is around 60% [17]. These statistics, although they provide evidence of good response to currently available treatments, strongly point to the need for further improvements (ideally, we should have a 5-year survival close to 100%). The major side-effects of cervical cancer chemotherapy include nausea and vomiting, loss of appetite, hair loss, mouth sores, and fatigue [18]. Finding novel chemicals with strong anticancer properties and reduced side effects is highly desirable.

As mentioned above, paclitaxel is used in the treatment of cervical cancer. Paclitaxel is derived from the bark of the Pacific yew tree (*Taxus brevifolia*) and represents a good example of a plant-derived compound developed into an effective chemotherapy agent [19]. The search for novel plant-derived chemicals that can counteract cancer is on-going and hopefully will result in new, more effective, and with fewer side-effects chemotherapy agents.

### 1.2. Resveratrol

Resveratrol (3,4′,5-trihydroxy-trans-stilbene) (Figure 1) is a polyphenolic stilbenoid first isolated from the roots of white hellebore (*Veratrum grandiflorum*) and Japanese knotweed (*Polygonum cuspidatum*) [20,21], which remains its dominant source, and is found in high concentration in grapes, blueberries, plums, apples, peanuts and red wine [22]. Resveratrol (RSV) was first isolated from *Veratrum grandiflorum* in 1939 by Takaoka et al. [23] and has extensively been investigated with over 20,000 research papers (including in vitro and in vivo studies) published today.

RSV has been shown to have an array of biological effects, including antioxidant [24], anti-inflammatory [25], neuroprotective [26], anti-diabetic [27], and anti-cancer properties [28,29,30,31]. The first study to examine the anticancer properties of RSV was published in 1997 by John Pezzuto’s group [30].

Although a number of reviews exist focusing on the anticancer effects of RSV [28,29,30,31,32,33,34,35], none are focused on cervical cancer. The current review summarizes all the in vitro and in vivo studies that examined the effects of RSV and its analogs on cervical cancer.

## 2. Resveratrol against Cervical Cancer

A number of studies have examined the effects of RSV against cervical cancer utilizing in vitro cell culture and in vivo xenograft animal models.

### 2.1. Resveratrol against Cervical Cancer: In Vitro Studies

Studies that have examined the effects of RSV on different cervical cancer cells in vitro are presented below and in Table 1 in chronological order. The first study was published in 2002 and the most resent one was published in 2022.

Using HeLa and SiHa cervical cancer cells, Zoberi et al. found that RSV enhanced the response to ionizing radiation treatment [36]. Cell growth and survival were significantly reduced in the presence of RSV compared to irradiated cells without RSV. This radiosensitizing effect was associated with a significant cell cycle arrest at the S phase and a significant inhibition of cyclooxygenase-1 (COX-1) activity [36].

Treatment of HeLa cells with RSV resulted in the inhibition of cell migration and invasion [37]. Furthermore, RSV treatment significantly reduced the phorbol myristate acetate (PMA)-induced matrix metalloproteinase-9 (MMP-9) mRNA, protein, and activity levels. This inhibition was associated with an inhibition of PMA-induced c-Jun *N*-terminal kinase (JNK) and protein kinase C (PKCδ)-activation [37]. In addition, RSV significantly reduced activator protein-1 (AP1) and nuclear factor-kappa B (NFkB) signaling [37].

Kramer et al. found that treatment of HeLa cells with RSV significantly reduced their proliferation and induced cell cycle arrest at the S-phase (observed 24 h after RSV treatment) [38]. However, this cell cycle arrest seemed to be reversible as the G2/M-cell population reappeared after 32–48 h of RSV treatment. Overall, this study shows that RSV has the ability to induce the S-phase arrest in HeLa cells but in a transient and reversible way [38].

Tang et al. found that human cervical cancer *C*-33A and HeLa cells transfected with the HPV-16 oncoproteins E6 and E7 had increased expression of HIF-1a and VEGF [39]. The phosphatidylinositol-3-kinase (PI3K) and the ERK signaling cascades were involved in this process, as seen by the increased levels of phosphorylated Akt and ERK. The use of specific inhibitors of these cascades (LY294002 to block PI3K and PD98059 to block ERK) significantly reduced the E6 and E7-induced HIF-1a levels. Importantly, treatment with RSV dose-dependently reduced the E6 and E7-induced HIF-1a and VEGF levels. Treatment with RSV had the same effect on HIF-1a levels as treatment with HIF-1a siRNA. This study shows that RSV has a potent anti-angiogenic effect and inhibits HPV E6 and E7-driven cervical cancer.

Hsu et al. found that treatment of HeLa, Cx, SiHa, and SKGIIIb cervical cancer cells with RSV (100–200 μΜ) resulted in inhibition of growth and induction of apoptosis, as seen by the increase in cytochrome C and cleaved caspase-3 levels [40]. Furthermore, treatment of HeLa and Cx cells with RSV time-dependently increased autophagy, as seen by the increased levels of LC3-II, a marker of autophagy. In addition, treatment of these two cell lines with 100 μΜ RSV resulted in increased cytosolic levels and activity of cathepsin L (cat L). Employment of a small interfering RNA (siRNA) approach to down-regulate cat L abolished the RSV-induced apoptosis while siRNA down regulation of squamous cell carcinoma antigen 1 (SCCA1) promoted RSV-induced apoptosis indicating a role as a negative regulator. Treatment with RSV increased lysosomal membrane permeability (LMP). Inhibition of autophagy with wortmannin or asparagine attenuated the RSV-induced responses (decreased LMP, LC3-II, and apoptosis). Overall, this study indicates that RSV induces autophagy resulting in a downstream increase in LMP, cytosolic cathepsin L, cytochrome C, and cleaved caspase-3 and apoptosis of cervical cancer cells [40].

Treatment of several cervical cancer cells (SKG-I, SKG-II, SKG-IIIa, Nuz & HeLa) with RSV (10, 30 & 100 μΜ) resulted in reduced survival that was associated with a reduction in the anti-autophagy factor ATPase family AAA domain containing 3A (ATAD3A) expression [41]. In addition, treatment of these cells with RSV increased autophagy and apoptosis. ATAD3A is increased in HPV-induced cervical cancer, and this study shows that RSV has the potential to counteract HPV-driven processes.

Kim et al. treated HeLa cells with RSV (10, 30 & 100 μΜ) and found a significant inhibition of phorbol 12-myristate 13-acetate (PMA)-induced cell migration and invasion [42]. This effect was associated with a decrease in matrix metalloproteinase-9 (MMP-9) levels and activity, NF-κB, and AP-1. In addition, RSV inhibited NF-κB mediated MMP-9 transcription. Overall, this study indicates that RSV has the potential to inhibit metastasis of cervical cancer [42].

Treatment of HeLa cervical cancer cells with RSV (25 μΜ) resulted in increased apoptosis, as was evident by the increased cleaved caspase-3 and caspase-9 levels, reduced mitochondrial membrane potential, and increased DNA fragmentation. [43]. A decrease in the level of HDM2 gene expression was detected after RSV treatment. Overall, this study shows that RSV has the ability to cause apoptosis in cervical cancer cells through the activation of the caspase cascade and by lowering the expression of HDM2 [43].

García-Zepeda et al. treated several cervical cancer cell lines (C33A, HeLa, CaLo, CaSki & SiHa), including HPV 16 and HPV18 positive ones, with RSV (150–250 μM for 48 h) and found a significant reduction in cell proliferation and induction of apoptosis and autophagy [44]. In addition, the majority of the cell lines showed an increased expression of p53, while the expression of p65 NFkB was reduced by RSV treatment.

HeLa and SiHa cervical cancer cells treated with RSV (100 µM for 48 h) had reduced growth and increased apoptosis. Western blot analysis revealed decreased phosphorylation of Notch1/2, Hes1, Wnt2/5a, β-catenin, and signal transducer and activator of transcription (STAT3) while the levels of PIAS3 were increased [45]. Treatment of the cells with RSV resulted in S-phase cell cycle arrest and induction of apoptosis, effects that were similar to the treatment with AG490, a selective STAT3/JAK3 inhibitor. The use of L-685,458, a Notch inhibitor, or XAV-939, a Wnt inhibitor, did not mimic the RSV-induced effects [45]. These data suggest that RSV-induced apoptosis is mediated by the inhibition of STAT3 [45].

Similar to these findings [45], Li et al. found that treatment of HeLa cervical cancer cells with RSV (10–100 μΜ) resulted in reduced viability, proliferation, and survival that was associated with a decrease in STAT3 phosphorylation and an increase in the expression of the gene associated with retinoid–IFN-induced mortality-19 (GRIM-19). Overexpression of GRIM-19 reduced *p*-STAT3, cyclin B1, VEGF, and Bcl-2 protein levels while downregulation of GRIM-19, using a siRNA approach, increased phosphorylation/activation of STAT3 and attenuated the effects of RSV. Overall, this study provides evidence that RSV targets the GRIM-19-STAT3 signaling cascade resulting in anticancer effects [46].

Zhang et al. treated three different cervical cancer cell lines (HeLa, SiHa & C33A) with 100 µM of RSV for 48 h and found a decrease in growth, proliferation, and induction of apoptosis. Immunocytochemical (ICC) staining, western blot, and RT-PCR analysis revealed a decrease in *p*-STAT3 and inhibition of survivin, c-Myc, cyclin D1, and VEGF expression in all three cell lines. Furthermore, basal levels of suppressor of cytokine signaling 3 (SOCS3) and protein inhibitor of activated STAT3 (PIAS3) were low and increased with RSV treatment, as seen by ICC staining and western blot analysis. Nuclear labeling revealed a negative correlation between PIAS3 expression and STAT3 nuclear translocation. These data suggest that RSV is able to modify STAT3 phosphorylation, activation, and nuclear translocation by increasing the expression of PIAS3 resulting in the inhibition of proliferation and survival of cervical cancer cells [47].

Ruíz et al. used HeLa cervical cancer cells enriched with cancer stem cells (CSC) positive for CD49f called HeLa Sphere (HeLa SP), which are less sensitive to Etoposide (VP16) [48]. Treatment of these cells with RSV and VP16 resulted in a significant decrease in cell viability and an increase in apoptosis that was associated with a reduction in RAD51 protein levels. Downregulation of RAD51, using siRNA, had similar effects as RSV treatment [48]. This study indicates that RSV has the ability to target RAD51 and increase the effectiveness of VP16 treatment in CSC-enriched cervical cancer cells.

Chatterjee et al. found that treatment of HeLa cervical cancer cells with RSV (5–40 µM for 24–48 h) reduced cell viability and cell migration and arrested the cell cycle to the S phase [49]. More importantly, RSV downregulated the HPV oncoprotein E6, induced caspase-3 activation, and upregulated p53 protein levels [49].

HeLa cervical cancer cells treated with RSV showed decreased proliferation (IC_50_: 21.76 µM), increased cell cycle arrest at the G1/S phase, and increased apoptosis. Western blot analysis revealed a significant increase in p53 protein levels with RSV treatment [50]. These effects of RSV were also seen in HeLa cells lacking the aryl hydrocarbon receptor (AHR), indicating that the RSV-induced effects are AHR-independent.

Exposure of HeLa cervical cancer cells to RSV reduced viability inhibited proliferation, and induced apoptosis in a dose- and time-dependent manner [51]. Treatment with RSV increased caspase-3 and -9 cleavages and increased the expression of the pro-apoptotic protein Bax whereas the levels of the antiapoptotic proteins Bcl-2 and Bcl-XL were decreased. Lastly, p53 expression levels were increased, and Cyclin B1 expression levels were decreased [51]. Overall, this study shows that RSV has a strong inhibitory effect on HeLa cells.

Treatment of HeLa cervical cancer cells with RSV reduced viability and proliferation was correlated with the downregulation of phospholipid scramblase 1 (PLSCR1) [52].

SiHa cervical cancer cells treated with RSV (100 µM for 24 h) showed reduced expression of survivin, an anti-apoptotic protein, which is often over-expressed in cervical cancer cells [53]. Knocking down of survivin using siRNA resulted in effects similar to RSV treatment. Furthermore, when SiHa cells were treated with RSV in combination with tumor necrosis factor-related apoptosis-inducing ligand (TRAIL), there was a notable decrease in cell viability, cell cycle arrest at the G2/M phase, increased apoptosis, and upregulation of E-cadherin. These data suggest that RSV, in combination with TRAIL, has a synergistic effect on inducing apoptosis in SiHa cancer cells [53].

Assad et al., treated HeLa cells with RSV and found reduced viability and cytotoxicity similar to that seen when cells were treated with the mTOR inhibitors Temsirolimus and Everolimus used in cancer treatment [54]. Furthermore, treatment with RSV, similar to treatment with Temsirolimus and Everolimus, enhanced the cytotoxic effect of radiation treatment (2Gy) [54]. This study indicates radio-sensitizing properties of RSV.

Treatment of HeLa and SiHa, human cervical cancer cells with RSV, resulted in reduced proliferation, reduced wound healing, migration, and invasion, indicating a reduction in metastasis [56]. Treatment with RSV (40 µM for 24 h) resulted in a significant increase in E-cadherin and a reduction in *N*-cadherin and vimentin, suggesting inhibition of epithelial to mesenchymal transition (EMT). Further supporting the decrease in the invasion, western blot analysis revealed a decrease in MMP-3 and -13 protein levels with RSV treatment [56]. In addition, treatment with RSV reduced the phosphorylation/activation of STAT3 protein [56].

HeLa cells treated with RSV showed reduced proliferation and increased apoptosis, as revealed by both flow cytometry/annexin measurements and the increased levels of the pro-apoptotic protein BIM [55]. Western blot analysis revealed reduced phosphorylated ERK and FOXO3a levels with RSV treatment [55].

Pani et al. found that treatment of HeLa cells with RSV resulted in reduced viability (MTT assay) that was associated with a decrease in intracellular glucose, suggesting a decrease in glucose uptake by the cancer cells [57]. Similarly, lactate levels were shown to be decreased, whereas pyruvate levels were increased, and the NADH/NAD^+^ ratio was decreased with RSV treatment suggesting RSV has effects that combat the Warburg Effect [57].

Treatment of HeLa and Ca Ski cells with RSV inhibited proliferation, induced cell cycle arrest, and promoted apoptosis. Flow cytometry results revealed an increased proportion of cells in the G1 phase of the cell cycle, while there was a significant decrease in the proportion of cells in the S phase with RSV treatment. The mRNA levels of p16, p21, and p27 were increased, whereas protein levels of CDK4, E2F1 and *p*-pRb1 were decreased. The mRNA and protein levels of BCL-2 were decreased, while mRNA and protein levels of BAX were increased with RSV treatment. In addition, immunocytochemistry and western blot analysis revealed a significant decrease in E6, E7, and *p*-pRb1 and an increase in p53 [59].

HT-3 cervical cancer cells treated with RSV had reduced viability and proliferation and increased apoptosis [60]. Maximum induction of apoptosis was seen at a concentration of 1.25 µM RSV. Unfortunately, this study did not examine the mechanisms involved in mediating these effects of RSV.

Treatment with HeLa cells with RSV resulted in reduced viability [61] and increased the expression (mRNA levels measured) of the sodium/lithium/calcium exchanger (NCLX). NCLX is a mitochondria membrane protein moving calcium out of mitochondria (into cytoplasm) in exchange for sodium or lithium. The increased expression of NCLX seen with RSV treatment correlated with increased cytosolic calcium levels. These data suggest that the reduced cell viability seen with RSV is due to calcium-induced cytotoxicity. The effect of RSV on cell viability was similar to that seen with the downregulation of NCLX by the siRNA approach [61]. Downregulation of NCLX leads to the overloading of the mitochondria with calcium resulting in cytotoxicity. The combination of RSV and NCLX siRNA resulted in a greater reduction in cell viability compared to each treatment alone. Overall, this study indicates that disruption of intracellular calcium homeostasis leads to cytotoxicity, and a combination of RSV and NCLX siRNA treatment approach should be further explored as an anticancer treatment.

When subclones of W12 cells (derived from cervical precancerous lesions containing episomal and integrated HPV16 DNA) were treated with RSV, their proliferation was significantly reduced. The IC_50_ RSV values in 20,861, 20,822, 201,402, 20,862, and 20,850 subclones of cells were in the range of 10–53.7 µM. [58].

Treatment of HeLa cells with RSV resulted in reduced survival and cell cycle arrest at the S-phase [62]. Immunofluorescence imaging revealed a significant downregulation of epidermal growth factor receptor (EGFR) in HeLa cells treated with RSV [62].

Although the in vitro studies presented above and in Table 1 utilized RSV concentrations in the range of 0.16 to 262 µM, the majority of the studies found strong anticancer effects with 50–100 µM RSV. Unfortunately, none of the studies summarized here utilized non-cancerous human cervical epithelial cells to examine the effects of RSV on cells representing normal/healthy cells. We performed a further search of the literature and found no studies examining the effects of RSV on healthy cervical epithelial cells. However, in the past, we found a significant inhibition of survival of PC3 and 22RV1 prostate cancer cells by RSV treatment, while PNT1A cells representing normal prostate epithelial cells were not affected [63].

Overall, the in vitro studies described in the above section provide evidence that RSV has the potential to inhibit proliferation and survival and induce apoptosis of cervical cancer cells (Table 1). Evidence regarding the mechanisms involved in these anticancer effects of RSV reveals inhibition of EGFR, ERK, JNK, PKC, STAT3, and NF-kB. The levels of the tumor suppressor p53 and the pro-apoptotic cleaved caspases-3, -8, and -9 and PARP were increased, while the levels of the anti-apoptotic Bcl-2 and Bcl-XL were reduced. Furthermore, RSV increased LC3-II levels indicating induction of autophagy, while reduced MMP 3, 9, and 13 levels reveal a potential to reduce metastasis and invasion (Figure 2).

### 2.2. Resveratrol Analogs against Cervical Cancer: In Vitro

RSV is a molecule with low bioavailability (discussed at the end of Section 2.3), and new analogs are being developed with the goal of increasing bioavailability as well as effectiveness while decreasing potential toxicity to healthy/normal tissues.

A number of analogs of RSV have been synthesized, and their anticancer potential has been examined utilizing cervical cancer cells and summarized in Table 2. Certain RSV analogs were found to be more effective than the parent compound of RSV. It should be noted that a comparison with the parent RSV compound was not performed in all studies.

Kim et al. examined the effects of nine synthetic analogs of RSV (styryl quinazoline derivatives) and found that (E)-8-acetoxy2-[2-(3,4-diacetoxyphenyl) ethenyl]-quinazoline (8-ADEQ) was the most potent in inhibiting the proliferation of HeLa cells (Table 2) Further examination of the mechanism involved revealed induction of cell cycle arrest at the G2/M phase that was associated with increased cyclin B1 expression, cyclin-dependent protein kinase 1 (Cdk1), cell division cycle 25C (Cdc25C) and p53 phosphorylation, checkpoint kinases 1 (Chk1) and Chk2 activation as well as activation of the ataxia telangiectasia mutated (ATM)/ataxia telangiectasia-Rad3-related (ATR) kinases. Inhibition of the ATM/ATR pathway using the inhibitor caffeine, or a siRNA approach attenuated the 8-ADEQ-induced G2/M cell cycle arrest. These data indicate that the RSV analog 8-ADEQ activates the ATM/ATR-p53 pathway leading to the inhibition of cervical cancer cell proliferation [64].

HeLa cells treated with the RSV analog pterostilbene had altered cell morphology, reduced proliferation, increased DNA fragmentation, and induced apoptosis, as indicated by the reduced mitochondrial membrane potential. Western blot analysis revealed a dose-dependent decrease in phosphorylation (activation) of mTOR and PI3K/Akt, which are implicated in cell proliferation and survival [65] (Table 2).

HeLa cells treated with *N*-(4-methoxyphenyl)-3,5-dimethoxybenamide (MPDB), an RSV analog, showed reduced proliferation and survival and cell cycle arrest at G2/M phase while p53 phosphorylation levels were elevated [66]. An increase in phosphorylation of Cdc2, Cdc25c, and Chk1/2 was seen. Treatment with MPDB reduced total and phosphorylated levels of the antiapoptotic protein Bcl-xL, increased cleavage of caspases-3, -8, and -9 and PARP, and increased phosphorylation of the Fas death receptor. The use of the caspase inhibitor, z-VAD-fmk (50 µM), abolished the MPDB-induced DNA fragmentation [66]. These data indicate that MPDB induced apoptosis by altering both the intrinsic and extrinsic pathways of apoptosis.

Both RSV and pterostilbene showed a significant decrease in TC1 cell viability. However, pterostilbene was found to be more potent, with an IC_50_ value of 15.61 µM compared to the parent RSV compound, with an IC_50_ value of 34.46 µM. Immunocytochemical analysis revealed a significant downregulation of E6 oncoprotein with both RSV and pterostilbene treatment of TC1 cells [67].

HeLa cells treated with pterostilbene had decreased growth, survival, and metastatic potential. A decrease in cell invasion and migration was seen through a wound-healing assay [68]. Both RSV and pterostilbene caused cell cycle arrest at the S and G2/M phases. The levels of tumor suppressor proteins p21 and p53 were significantly increased while the expression of cyclin E1/B1 was reduced. An increase in cleavage of caspase-3 and -9 and a decrease in the anti-apoptotic protein Bcl-2 and Bcl-XL were seen. Matrix metalloproteinases (MMPs)-2/-9, which are implicated in cancer metastasis, were significantly reduced with pterostilbene treatment [68].

### 2.3. Resveratrol against Cervical Cancer: In Vivo Animal Studies

Few studies have examined the effects of RSV administration in animal models of cervical cancer which are presented below and in Table 3.

BALB/C nude mice were subcutaneously injected with HeLa cells and were treated with RSV (10 mg/kg of body weight) daily for 28 days. RSV treatment resulted in a significant reduction in tumor weight [52]. Unfortunately, no other measurements or tumor analysis was performed in this study.

Female C57BL/6 mice were subcutaneously injected with TC-1 cells (expressing the oncogenes: HPV16-E6, -E7, and H-Ras) in the nape. After tumors were well established (15 to 20 days following cancer cell injection), tumors were sectioned into four quadrants, and each quadrant was treated with RSV (10 μL of 1 mM stock solution, a total of 40 μL injected intralesionally) for 5 consecutive days. Although the concentration of RSV within the tumor was not measured, it was expected to be above the lethal dose of 80 µM. This treatment resulted in significantly smaller tumor size and volume compared to animals treated with vehicle control (PBS). Immunohistochemistry of the tumor tissue revealed a significant reduction in E6 oncoprotein levels, vascular endothelial growth factor (VEGF), and proliferating cellular nuclear antigen (PCNA) of 79%, 60.5%, and 75.5%, respectively. Taken together, these data indicate that direct treatment of tumors with RSV resulted in significant tumor shrinkage that was due to inhibition of proliferation and cell cycle arrest associated with a reduction in E6 protein levels [67].

Athymic BALB/C nude mice were divided into an RSV pre-treatment and treatment group [56]. The pretreated group was administered intragastrically with 30 mg/kg of RSV 3 times per week for two weeks before being subcutaneously injected with HeLa cells (5 × 10^6^ cells), then treated with RSV for an additional 3 weeks. The treated group was administered intragastrically with RSV (30 mg/kg, 3 times/week) 10 days after HeLa cell injection for a total of 5 weeks. The data from the RSV pre-treatment and treatment groups illustrated a significant decrease in tumor size, weight, and volume. Western blot analysis of the tumor samples showed a significant reduction in STAT3 phosphorylation, b-catenin, vimentin, MMP-3, and MMP-13, levels. IHC also revealed it decreased in *p*-STAT3, *N*-cadherin, and vimentin and increased E-cadherin [56]. Unfortunately, with this experimental design of the study, it is not clear what is the effect of the RSV pretreatment alone. The researchers should have included another group of animals that received cervical cancer cells pretreated with RSV and not treated with RSV for the rest of the study.

Antitumor effects were also witnessed in another study conducted by Hao et al. BALB/C female nude mice subcutaneously injected with 2 × 10^6^ HeLa cells and 1 week later, were orally treated with 30 mg/kg RSV, three times per week, for 3 weeks. This treatment resulted in decreased tumor volume and weight compared to the control. mRNA and protein levels of both E6 and E7 were found to be reduced, while protein levels (measured by western blotting and immunostaining) of tumor suppressors p53 and Rb1 were significantly increased in tumor tissues [69].

Nude Balb/C female mice subcutaneously injected with HeLa cells were intragastrically administered 15 mg/kg of RSV three times a week for 5 consecutive weeks. With this treatment course, tumor weight and volume decreased significantly. Western blot analysis of the tumor tissue revealed a decrease in E6 and E7, and *p*-pRb1 with an increase in the tumor suppressor p53 [59].

Although the above-described in vivo animal studies show that administration of RSV in mice xenografted with cervical cancer cells resulted in a significant reduction in tumor volume and weight compared to non-treated mice (Figure 3), unfortunately, none of the studies measured blood RSV levels. The absorption of RSV through the gastrointestinal tract is low, and this results in low bioavailability [70]. In addition, RSV is rapidly metabolized and eliminated by systemic circulation, further contributing to its low bioavailability [70]. Intragastric administration of RSV (472 mg) in pigs, followed by an examination of RSV and its metabolites, showed dihydro-resveratrol (DHR) and RSV-3-O-glucuronide as the main metabolites [71]. A recent study has found that oral (in the diet) administration of RSV in mice for 4 weeks (human equivalent dose of 4.6 mg/kg/day) resulted in the detection by HPLC-MS/MS of 11 RSV metabolites in tissues (bile, kidney, liver) and plasma that included dihydro-resveratrol (DHR), lunularin (LUN), and sulfate and glucuronide conjugates of RSV [72]. In plasma DHR, LUN, and RSV conjugates of 2–7 µM were found. Importantly, the evidence indicates that DHR and LUN are gut bacteria-derived metabolites of RSV, as these metabolites were not detected in antibiotic-treated mice. Furthermore, DHR and LUN, at the concentrations found in mouse tissues, had stronger anti-inflammatory and anti-cancer effects than RSV. This study provides evidence of the important role of gut microbiota on RSV metabolism. Importantly, the gut microbiota-generated RSV metabolites appear to significantly contribute to the biological effects of RSV in vivo.

Limited studies exist investigating the bioavailability of RSV in humans. In one study, single oral administration of 25 mg RSV in humans resulted in blood levels of 2 µM after 60 min [73], while in another study, oral administration of 500 to 5000 mg RSV resulted in plasma levels of 0.3–2.3 µM after 50–90 min [74]. The above studies indicate that oral administration of RSV, despite its low bioavailability, results in plasma RSV and RSV metabolite levels in the micromolar range [71,72,73,74].

In the U.S.A., RSV is sold as a dietary supplement in health food stores, and this does not require Food and Drug Administration’s (FDA) approval. As per the Federal Food, Drug, and Cosmetic Act (FD&C Act) which was amended in 1994, the FDA does not have the authority to approve any dietary supplements but rather regulate the product after it enters the marketplace. The capsules sold in health food stores contain 200–800 mg RSV, and the recommended daily dose of RSV is around 400–800 mg and to not exceed 1000 mg. Hopefully, more clinical studies will be performed in the future to determine the optimal dose of RSV for treating different diseases, including cervical cancer.

## 3. Limitations and Controversies

From the in vitro studies, it is evident that the effects of RSV on cells representing non-cancerous human cervical tissue have not been examined, and further investigation is needed. Future in vitro studies should examine in parallel the effects of RSV on both cancer and non-cancer cervical cells. Primary human cervical epithelial cells such as the PCS-480–011 from ATCC could be used in such studies.

Another limitation of the in vitro studies is the lack of use of three-dimensional (3-D) cervical cancer models [75]. Tumor cells in vivo exist in a tumor microenvironment (TME) composed of extracellular matrix (ECM), supporting stromal cells, blood vasculature, and infiltrating immune cells. The interaction between tumor cells and TME plays a pivotal role in tumor development and progression. The use of 3D in vitro tumor models [75] mimics the in vivo TME better and, therefore, will provide significant data on the potential inhibitory effects of RSV against cervical cancer.

Although we found many (a total of 27) studies examining the effects of RSV in cervical cancer cells in vitro, unfortunately, only five animal studies exist examining the in vivo effects of RSV. All these animal studies utilized ectopic xenograft models of cervical cancer. In these models, immunosuppressed mice are subcutaneously injected with human cervical cancer cells, typically on their flank. Although these animal models are easy to establish and manage and provide a good quantitation of the tumor burden, they have disadvantages, as they do not mimic local tumor growth and they do not recapitulate the tumor microenvironment existing in the cervix of patients with cervical cancer. The use of orthotopic models [76] of cervical cancer is advantageous and recapitulates the tumor microenvironment and the signaling pathways involved in tumor growth and metastasis seen in human cervical cancers. Hopefully, future studies will utilize orthotopic and clinically relevant mouse models (surgical orthotopic implantation of cells derived from tumors excised from the cervix of patients) to study the effects of RSV.

Another major limitation in our understanding of the effects of RSV against cervical cancer is the lack of clinical studies. We found 191 clinical trials related to RSV (we searched ClinicalTrials.gov), 17 of which are related to cancer. Unfortunately, no trials exist with RSV and cervical cancer. Among the 17 cancer-related studies, one has reported results. This study (now terminated) (NCT00920556) examined the safety and activity of RSV (SRT501) alone or in combination with Bortezomib in 24 patients with multiple myeloma. Participants were given 5.0 g of RSV through oral suspension every morning for 20 days for a maximum period of 12 cycles, and unfortunately, 12 out of the 24 participants had serious adverse effects. The dose of 5 g of RSV is much higher than in other clinical/metabolic studies [77,78]. Another completed study with no reported results (NCT00256334) examined the effect of RSV on 11 patients with colon cancer who were given either 20, 80, or 160 mg of RSV per day. Overall, the existing clinical trials related to RSV administration in cancer patients have large discrepancies in the dose of RSV used (5 g in the NCT00920556 versus 20–160 mg in the NCT00256334 trial). It should be noted that in other trials examining the effects of RSV in patients with diabetes [77,78], the doses of RSV range from 150 mg to a maximum of 1.5 g. Hopefully, these dose discrepancies will be addressed in future clinical trials.

Likely, if more animal studies are performed in the future and provide strong evidence of the anticancer effects of RSV, clinical studies will be performed as well.

It should be noted that the structure of RSV resembles the structure of the steroid female sex hormone estrogen, and for this reason, RSV is referred to as phytoestrogen. Many studies utilizing breast cancer cells in vitro and animal models of breast cancer have shown RSV to act as an estrogen receptor antagonist and induce anticancer effects [79,80]. However, studies have also reported the potential estrogenic activity of RSV. Gehm et al. reported in 1997 that RSV (3–10 µM) inhibited the binding of estradiol to its receptor and increased the transcription of estrogen-responsive reporter genes in human breast cancer cells, including MCF-7 cells [81]. This effect was estrogen receptor-dependent and was inhibited by specific estrogen antagonists. Overall, this was the first study indicating that RSV acts as an estrogen agonist. In another study, Bowers et al. found that RSV binds to both estrogen receptor alpha (ERα) and beta (ERβ) and acted as an estrogen agonist, as seen by the increased ERE-driven reporter gene activity in CHO-K1 cells expressing either receptor [82]. In another study, the treatment of breast cancer MCF-7 cells with RSV was found to induce apoptosis by a mechanism involving ERK activation and p53 phosphorylation and acetylation, indicating anticancer effects that are p53-dependent [83]. Interestingly, these effects of RSV were blocked in the presence of estrogen [83]. The above studies were performed utilizing in vitro models, and unfortunately, the estrogenic potential of RSV in animals and humans is unknown. In a pilot study conducted in postmenopausal women with high body mass index (BMI ≥ 25 kg/m^2^), administration of 1 g of RSV daily for 12 weeks did not result in significant changes in serum concentrations of estradiol, estrone, and testosterone but led to an average of 10% increase in the concentrations of sex steroid hormone binding globulin (SHBG) [84]. This elevation in SHBG could reduce the bioavailability and, thus, the bioactivity of sex steroid hormones, including estrogen. Clearly, more animal and clinical studies are required to examine whether RSV acts as an estrogen receptor agonist or antagonist in vivo.

RSV is established as an activator of silent information regulator-1 (SIRT1), a class III histone protein deacetylase (HDACs) involved in many cellular processes [85]. Treatment of colorectal cancer cells with RSV increased SIRT1 activation, decreased proliferation, invasion/metastasis, and increased mesenchymal to epithelial transition (MET) [86]. Similarly, RSV activated SIRT1, reduced NFkB, and acted against alcohol-aflatoxin B1-induced Hepatocellular carcinoma (HCC)[87].

However, the role of SIRT-1 in cervical cancer is not clear. Wang et al. found that expression of SIRT1 (measured by immunohistochemistry) in cervical tumor tissues was increased with cancer progression and a positive correlation existed between SIRT1 and HPV 16 E7 protein levels. In in vitro studies silencing of HPV16 E7 reduced SIRT1 levels. SIRT1 depletion inhibited proliferation, migration, and invasion and induced apoptosis of SiHa cervical cancer cells [88]. In a study by Velez -Perez et al., SIRT1 expression was also correlated with the progression of cervical squamous cell carcinoma (SCC) [89]. Although these studies [88,89] suggest that SIRT1 may serve as a biomarker for predicting cervical cancer progression, it is not clear if the increased SIRT1 levels are part of a compensatory mechanism attempting to counteract cancer progression.

A few review manuscripts are focused on RSV toxicity and adverse effects in humans [90,91,92]. Cottart et al. reviewed the existing studies up to 2010 and reported that oral administration of RSV in humans is overall well tolerated. It should be noted, however, that in the majority of these studies, RSV was administered as a single dose of less than 1 g in healthy subjects. Administration of RSV at higher doses and/or patients suffering from different conditions may result in different outcomes. Indeed, side effects such as nausea, vomiting, diarrhea, and liver dysfunction were seen in patients with non-alcoholic fatty liver disease at doses of 2.5 g or more per day [93]. Likewise, in a study by Pollack et al. [94], administration of 3 g RSV/daily to older glucose-intolerant adults for 6 weeks resulted in severe gastrointestinal symptoms that disappeared when the dose was reduced to 1 g twice daily. Overall, these data indicate that the reported gastrointestinal side effects of RSV are dose-dependent. Possible toxicity and adverse effects of long-term (years) RSV administration remains to be examined.

## 4. Conclusions

The evidence provided by the in vitro studies, summarised in this review manuscript, strongly indicates that treatment of cervical cancer cells with RSV reduces viability and proliferation while inducing apoptosis and autophagy. Some studies showed cell cycle arrest in the G1/S phase. Most studies reported induction of apoptosis seen by increased levels of cleaved PARP, caspases-3, -8, and -9, increased expression of the proapoptotic protein Bax and a significant decrease in the expression of the antiapoptotic proteins Bcl-2 and Bcl-XL. Apart from apoptosis, induction of autophagy was seen in some studies. In addition, treatment with RSV resulted in the inhibition of EGFR, VEGFR, PKC, JNK, ERK, NF-kB, and STAT3. Importantly, many studies have shown a reduction in the HPV oncoproteins E6 and E7 and increased levels of the tumor suppressor p53 with RSV treatment.

Animal studies have shown that treatment of mice, currying ectopic xenografted cervical cancer tumors, with RSV resulted in a decrease in tumor volume and weight.

A few RSV analogs have been created and investigated for their effectiveness against cervical cancer, with some showing higher anticancer effects than the parent RSV compound.

Overall, it is clear from the papers included in this review that treatment of cervical cancer cells with RSV significantly lowers essential characteristics of the disease, including cell viability, proliferation, metastasis, and invasion, in addition to causing programed cell death. RSV treatment decreased tumor volume and weight in animal models of cervical cancer. Future studies are needed to clarify the cellular signaling mechanisms involved and to fully understand the impact of RSV on both malignant and healthy tissues.

Importantly, more animal studies utilizing orthotopic and clinically relevant cervical cancer models and clinical trials are essential to better understand the anticancer potential of RSV in patients with cervical cancer.

## Figures and Tables

**Figure 1 nutrients-14-05273-f001:**
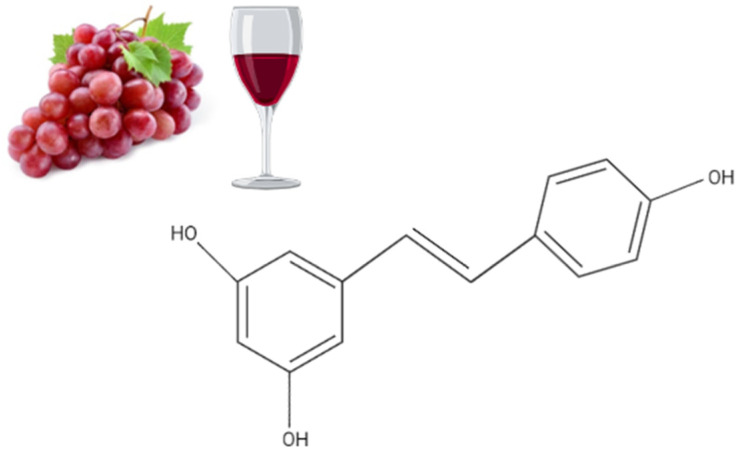
Chemical structure of resveratrol, a polyphenolic stilbene found in high concentration in grapes and red wine.

**Figure 2 nutrients-14-05273-f002:**
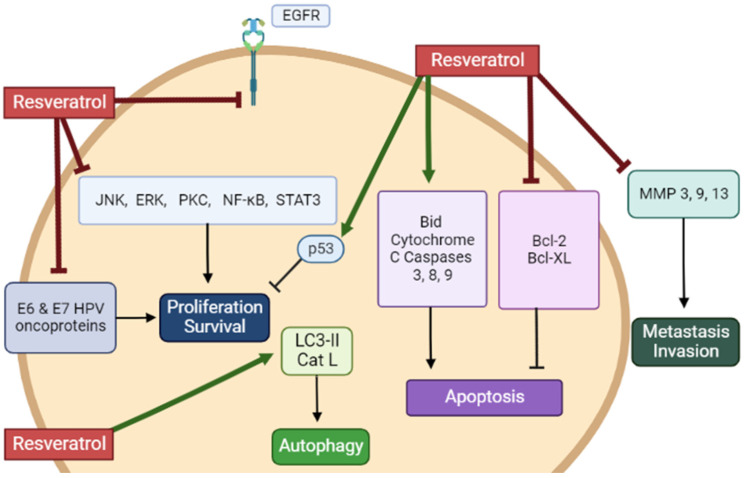
Summary of the effects of resveratrol in cervical cancer cells in vitro. RSV reduced proliferation and survival and induced apoptosis of cervical cancer cells. The figure, created using BioRender.com, is based on the data of the studies [36,37,38,39,40,41,42,43,44,45,46,47,48,49,50,51,52,53,54,55,56,57,58,59,60,61,62].

**Figure 3 nutrients-14-05273-f003:**
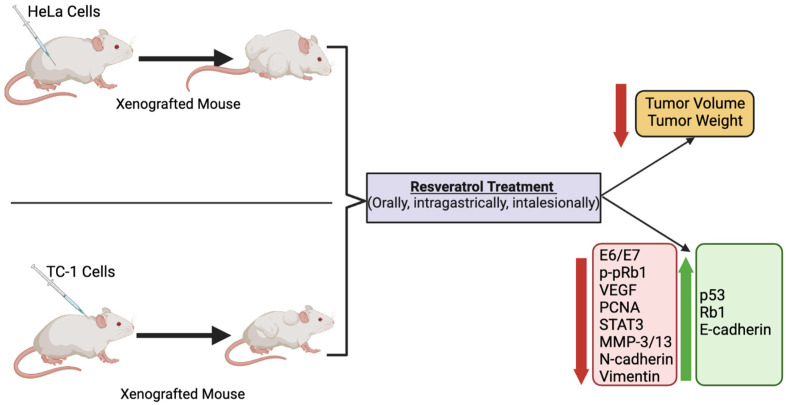
Summary of the effects of resveratrol in cervical cancer in animals. Treatment of animals (mice) xenografted with cervical cancer cells with RSV resulted in a significant reduction in tumor volume and weight compared to non-treated mice. The figure, created using BioRender.com, is based on the data of the studies [52,56,59,67,69].

**Table 1 nutrients-14-05273-t001:** Effects of Resveratrol against Cervical Cancer: in vitro studies.

Cell	Resveratrol Concentration/Duration	Effect	Reference
HeLa, SiHa cervical cancer cells	10, 25 µM 1–8 days	Increased effects of IR ↓cell growth ↓cell survival ↑cell cycle arrest (S phase) ↓COX-1 activity	[36]
HeLa cells	50 & 75 µM 24 h	↓PMA effects ↓MMP-9 mRNA, protein & activity ↓JNK ↓PKC δ ↓AP-1 ↓NFkB	[37]
HeLa cervical cancer cells	5, 25 & 50 µM 24–48 h	↓cell growth Accumulation in the S phase of the cell cycle	[38]
*C*-33A and HeLa Expressing HPV E6, E7	25, 50 &100 µM 16 h	↓HIF-1a ↓VEGF	[39]
hHeLa, Cx, SiHa and SKGIIIb cervical cancer cells	100–400 µM 24–72 h	↓cell growth ↑autophagy ↑apoptosis ↑LC3-II ↑LMP ↑Cat L ↑Cytochrome C ↑caspase-3	[40]
SKG-I SKG-II SKG-IIIa Nuz HeLa Cervical cancer cells	10, 30 & 100 μΜ	↑autophagy ↑apoptosis ↓drug resistance ↓ATAD3A ↑abrasion of the mitochondrial outer membrane ↑autophagosomes	[41]
HeLa cervical cancer cells	10, 30 & 100 μΜ 24 h	↓invasion ↓metastasis ↓MMP9 levels-activity ↓NF κΒ ↓AP-1	[42]
HeLa cervical cancer cells	25 μΜ 24, 48 & 72 h	↓cell proliferation ↑apoptosis ↑caspase-9 ↑caspase-3 ↓mitochondrial membrane potential -JC-1 in monomeric form ↓HDM2	[43]
C33A (with mutated p53) HeLa(HPV18positive) CaLo(HPV18positive) CaSki(HPV16positive) SiHa(HPV16 positive)	150–250 µM 48 h	↓proliferation ↑apoptosis ↓mitochondrial membrane potential ↑mitochondrial and lysosomal permeability ↑p53 levels ↓p65 NF κB levels	[44]
HeLa SiHa cervical cancer cells	100 μΜ 12–48 h	↑S-phase cell cycle arrest ↑apoptosis ↓*p*-STAT3 ↓Notch1/2 ↓Hes1 ↓Wnt2/5a ↓β-catenin ↑PIAS3	[45]
HeLa cervical cancer cells	10 & 100 μΜ 24 h	↓cell viability ↓cell proliferation ↓cell survival ↑GRIM-19 ↓*p*-STAT3 ↓cyclin B1 ↓VEGF ↓Bcl-2	[46]
HeLa SiHa C33A cervical cancer cells	100 μΜ 12, 24, 36 & 48 h	↓cell growth ↓proliferation ↑apoptosis ↓*p*-STAT3 ↓survivin ↓c-Myc ↓cyclin D1 ↓VEGF ↑SOCS3 ↑PIAS3	[47]
cancer stem cells (CSC) from HeLa cultures (HeLa SP)	137 μM 48–72 h	↓cell viability ↑apoptosis ↓RAD51	[48]
HeLa cervical cancer cells	5–40 µM 24–48 h	↓cell viability ↓cell migration ↑cell cycle arrest (S phase) ↓viral oncogene E6 ↑p53 levels	[49]
HeLa	10–80 µM 12–36 h	↓cell proliferation ↑cell cycle arrest at G1/S phase ↑p53 levels ↑apoptosis	[50]
HeLa	10–40 µM 24–48 h	↓cell viability ↓cell proliferation ↑apoptosis ↑caspase-3 ↑caspase-9 ↑Bax ↓Bcl-2 ↓Bcl-XL ↑p53 ↓Cyclin B1	[51]
HeLa	0–100 µM 24–96 h	↓Cell growth ↓Cell viability ↓Proliferation ↓Phospholipid scramblase 1	[52]
SiHa	100 µM 24 h	↓Cell viability ↑Cell cycle arrest in G2/M ↑Apoptosis ↓Survivin mRNA levels ↓Survivin protein levels ↑E-cadherin	[53]
HeLa	2.5–150 µM 24–48 h	↓Cell viability ↑Cytotoxicity ↑necrosis	[54]
HeLa	0–80 µM 48 h	↓Proliferation ↑Apoptosis ↓*p*-FOXO3a ↑FOXO3a ↑Bim ↓*p*-ERK	[55]
HeLa SiHa	0–40 µM 24 h	↓Proliferation ↓Wound healing ↓Migration/invasion ↓Metastasis ↑E-cadherin ↓*N*-cadherin ↓vimentin ↓MMP-3/13 protein levels ↓STAT3 protein levels	[56]
HeLa	20 µM 24 h	↓Cell viability ↑Cytotoxicity ↓Glucose uptake ↓NADH/NAD^+^ ratio ↓Lactate ↑Pyruvate	[57]
W12	0–100 µM	↓Proliferation	[58]
HeLa Ca Ski	5–40 µM 24 h	↓Proliferation ↑Cell cycle arrest in S phase ↑Apoptosis ↑p16/21/27 ↓CDK4 ↓E2F1 ↓*p*-pRb1 ↓Bcl-2 mRNA & protein levels ↓Bcl-xL mRNA levels ↑Bax protein levels ↑E6/7 ↑p53	[59]
HT-3	0.16–1.25 µM 0–48 h	↓Cell viability ↓Cell growth ↓Proliferation ↑Apoptosis	[60]
HeLa	262.87 µM 24 h	↓Cell viability ↑Apoptosis ↑mRNA caspases-3/-8/-9 levels ↑NCLX	[61]
HeLa	20 µM 24 h	↑Cell cycle arrest in S phase ↓Colony formation ↓EGFR	[62]

**Table 2 nutrients-14-05273-t002:** Effects of resveratrol analogs against cervical cancer: in vitro evidence.

Cell	Analog Name	Resveratrol Concentration/Duration	Effect	Reference
HeLa	8-ADEQ	8 µM 25 h	↓proliferation ↑Cell cycle arrest at G2/M phase ↑cyclin B1 levels ↑Cdk1, Cdc25C phosphorylation ↑Chk1, Chk2 activation ↑ATM/ATR activation	[64]
HeLa	Pterostilbene	0–400 µM 24–48 h	↑Cell morphology ↓Cell growth ↑DNA fragmentation ↓Proliferation ↑Apoptosis ↓*p*-mTOR ↓*p*-PI3K ↓*p*-Akt	[65]
HeLa	*N*-(4-methoxyphenyl)-3,5-dimethoxybenamide (MPDB)	35 µM 15 h	↓Cell growth ↓Survival ↑Cell cycle arrest at G2/M phase ↓Proliferation ↑DNA fragmentation ↑Apoptosis ↑Cdc2 ↑Cdc25c ↑Chk1/2 ↑p53 ↓Bcl-xL ↑Fas ↑Caspases-3/-8/-9 ↑Cleaved PARP	[66]
TC1	Pterostilbene	20, 30 µM 48 h	↓Cell viability ↑Cytotoxicity ↑Apoptosis ↓E6	[67]
HeLa	Pterostilbene	20 µM 24–48 h	↓Cell growth ↓Survival ↓Metastasis ↑Cell cycle arrest at S and G2/M phase ↑p21/53 protein levels ↓Cyclin E1/B1 ↓Bcl-2 protein levels ↓Bcl-xL protein levels ↑Cleaved caspases-3/-9 ↓MMPs-2/-9	[68]

**Table 3 nutrients-14-05273-t003:** Effects of Resveratrol against Cervical Cancer: in vivo animal studies.

Cell	Resveratrol Concentration/Duration	Effect	Reference
Female BALB/C nude mice subcutaneously injected 2 × 10^6^ HeLa cells/mL (100 µL/mouse)	10 mg/kg RSV orally, daily for 28 days	↓Tumor weight ↑PLSCR1	[52]
C57BL/6 female mice injected with TC-1 (HPV oncogene E6, E7 positive) cells subcutaneously	Injection of RSV intralesionally administrated for 5 days	↓Tumor size ↑cell cycle arrest ↓Tumor E6 levels ↓Tumor VEGF levels ↓Tumor PCNA levels	[67]
Athymic BALB/C nude mice subcutaneously injected 5 × 10^6^ HeLa cells/mouse	30 mg/kg RSV intragastrically, 3 times/week for 2 weeks (pre-treatment)	↓Tumor volume ↓Tumor weight ↓STAT3 protein levels ↓MMP-3/13 protein levels ↑E-cadherin ↓*N*-cadherin protein levels ↓Vimentin protein levels	[56]
Female BALB/C nude mice subcutaneously injected 2 × 10^6^ HeLa cells/mouse	30 mg/kg RSV orally, 3 times/week for 3 weeks	↓Tumor volume ↓Tumor weight ↓E6/7 mRNA levels ↓E6/7 protein levels ↑p53 expression ↑Rb1 expression	[69]
Female BALB/C nude mice subcutaneously injected 2 × 10^6^ HeLa cells/mL (100 µL/mouse)	15 mg/kg RSV intragastrically, 3 times/week for 5 weeks	↓Tumor volume ↓Tumor weight ↓E6/7 mRNA levels ↓E6/7 protein levels ↓*p*-pRb1 ↑p53 expression	[59]

## Data Availability

Not applicable.

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
