# Peer review of "Resveratrol against Cervical Cancer: Evidence from In Vitro and In Vivo Studies"

_nutrients, 2022, doi:10.3390/nu14245273_

Round 1

Reviewer 1 Report

The review entitled “Resveratrol against cervical cancer: evidence from in vitro and in vivo studies” summarizes the effectiveness of resveratrol against cervical cancer in vitro and in vivo. The manuscript is well organized, but I would like to encourage the authors to do some improvements.

1. In table 1, authors list different concentration/duration of resveratrol according to the references, and resveratrol treatments in the same cervical cancer cells have so large discrepancies in the used dosage. Are high doses of resveratrol toxic to healthy cells as well, and are the treatments appropriate? The authors had better discuss this.

2. In part 2.2 and table 2, authors summarized resveratrol analogs against cervical cancer in vitro, but in order to maintain the consistency and integrity of the review, the topic of resveratrol analogs is suggested to revise and put in the discussion section.

3. Line 45:"in cervical cancer tumors."should be “---in cervical cancer”.

4. The content of line 98-103 and line 141-146 looks similar, so it is suggested to describe together. 

5. The units “μΜ” or “μmol/l and “h”or“hrs”are appeared in the text, please standardize the use of units. 

Author Response

Reviewers' comments

Reviewer 1

The review entitled “Resveratrol against cervical cancer: evidence from in vitro and in vivo studies” summarizes the effectiveness of resveratrol against cervical cancer in vitro and in vivo. The manuscript is well organized, but I would like to encourage the authors to do some improvements.

  1. In table 1, authors list different concentration/duration of resveratrol according to the references, and resveratrol treatments in the samecervical cancer cells have so large discrepancies in the used dosage. Are high doses of resveratrol toxic to healthy cells as well, and are the treatments appropriate? The authors had better discuss this.

We thank this Reviewer for the comments.

We have addressed this comment in the section 2.1(in vitro studies) and 4(Limitations). Please see the revised manuscript and below.

2.1

Although the in vitro studies presented above and in table 1 utilized RSV concentrations in the range of 0.16 to 262 µM, the majority of the studies found strong anticancer effects with 50-100 µM RSV. Unfortunately, none of the studies summarized here utilized non-cancerous human cervical epithelial cells to examine the effects of RSV on cells representing normal/healthy cells. We performed a further search of the literature and found no studies examining the effects of RSV on healthy cervical epithelial cells. However, in the past, we found a significant inhibition of survival of PC3 and 22RV1 prostate cancer cells by RSV treatment while PNT1A cells representing normal prostate epithelial cells were not affected [63]. 

4.

From the in vitro studies, it is evident that the effects of RSV on cells representing non-cancerous human cervical tissue have not been examined and further investigation is needed. Future in vitro studies should examine in parallel the effects of RSV on both cancer and non-cancer cervical cells.  Primary human cervical epithelial cells such the PCS-480-011 from ATCC could be used in such studies.

  1. In part 2.2 and table 2, authors summarized resveratrol analogs against cervical cancer in vitro, but in order to maintain the consistency and integrity of the review, the topic of resveratrol analogs is suggested to revise and put in the discussion section.

We thank the reviewer for this comment. The topic of resveratrol analogs includes the summary of 5 different studies, a table, and is around 2 pages long. Therefore, presenting this information not in the discussion but as a separate section (section 2.2) is very appropriate. In addition, having a separate section enhances the organization and quality of our manuscript.

  1. Line 45:"in cervical cancer tumors."should be “---in cervical cancer”.

Addressed

  1. The content of line 98-103 and line 141-146 looks similar, so it is suggested to describe together. 

Addressed

  1. The units “μΜ” or “μmol/l and “h”or“hrs”are appeared in the text, please standardize the use of units. 

Addressed. We have also edited resveratrol to RSV in the text.

Reviewer 2 Report

Matteo Nadile et al. attempted to compile the role of resveratrol against cervical cancer based on in vitro and 2 in vivo studies. The manuscript is well written, and the question posed by the authors is well defined. The manuscript adheres to the relevant standards for reporting. However, few suggestions to strengthen the manuscript.

The authors need to mention the bioavailability of resveratrol in the introduction. The absorption and Metabolism can also be included in the manuscript.

In ref 55, Devi and Raj reported that 262.87 μM of resveratrol exposed to 24 h in HeLa cells showed apoptotic activity. A recent study by Pani et al. reported that 20 μM of resveratrol for 24 hrs showed a cytotoxic effect. In both studies, they used the same cell line. What could be the reason for the observed differences?

Is resveratrol approved by the U.S. Food and Drug Administration? That information needs to be provided

What are the limitations of using resveratrol as a therapeutic agent? 

The estrogenic activity of resveratrol has been reported in a few papers. If that is the case, how the authors can suggest anticancer treatment

What are the adverse effects of resveratrol?

Author Response

Reviewer 2

Matteo Nadile et al. attempted to compile the role of resveratrol against cervical cancer based on in vitro and 2 in vivo studies. The manuscript is well written, and the question posed by the authors is well defined. The manuscript adheres to the relevant standards for reporting. However, few suggestions to strengthen the manuscript.

  1. The authors need to mention the bioavailability of resveratrol in the introduction. The absorption and Metabolism can also be included in the manuscript.

We thank the Reviewer for his/her comments.

We have added information regarding bioavailability of resveratrol, metabolism and tissue distribution. This information is included in the end of the in vivo section (most appropriate) of the revised manuscript and below:

Although the above-described in vivo animal studies show that administration of RSV in mice xenografted with cervical cancer cells resulted in significant reduction in tumor volume and weight compared to non-treated mice (Figure 3), unfortunately none of the studies measured blood RSV levels. The absorption of RSV through the gastrointestinal tract is low and this results in low bioavailability [69]. In addition, RSV is rapidly metabolized and eliminated by the systemic circulation further contributing to its low bioavailability [69].  Intragastric administration of RSV (472 mg) in pigs followed by examination of RSV and its metabolites showed dihydro-resveratrol (DHR) and RSV-3-O-glucuronide as the main metabolites [70].

A recent study has found that  oral (in the diet) administration of RSV in mice for 4 weeks (human equivalent dose of 4.6 mg/kg/day) resulted in  detection by HPLC-MS/MS of 11 RSV metabolites in tissues (bile, kidney, liver) and plasma that included dihydro-resveratrol (DHR), lunularin (LUN), and sulfate and glucuronide conjugates of RSV [71]. In plasma DHR, LUN, and RSV conjugates of 2-7 µM were found. Importantly, the evidence indicates that DHR and LUN are gut bacteria-derived metabolites of RSV, as these metabolites were not detected in antibiotic-treated mice.  Furthermore, DHR and LUN, at the concentrations found in mouse tissues, had stronger anti-inflammatory and anti-cancer effects than RSV. This study provides evidence of the important role of the gut microbiota on RSV metabolism. Importantly, the gut microbiota-generated RSV metabolites appear to significantly contribute to the biological effects of RSV in vivo.

Limited studies exist investigating the bioavailability of RSV in humans. In one study, single oral administration of 25 mg RSV in humans resulted in blood levels of 2 µM after 60 min [72]. while in another study oral administration of 500 to 5000 mg RSV resulted in plasma levels of 0.3–2.3 µM after 50–90 min [73]. The above studies indicate that oral administration of RSV, despite its low  bioavailability, results in plasma RSV and RSV metabolite levels in micromolar range [70–73].

  1. In ref 55, Devi and Raj reported that 262.87 μM of resveratrol exposed to 24 h in HeLa cells showed apoptotic activity. A recent study by Pani et al. reported that 20 μM of resveratrol for 24 hrs showed a cytotoxic effect. In both studies, they used the same cell line. What could be the reason for the observed differences?

Both studies show that RSV inhibits the proliferation of cervical cancer cells in a dose dependent manner. The calculated IC50 values are 291.3, 50.09, and 8.73 μΜ in HeLa cells for 24, 48, and 72 h (Devi and Raj) and 43,36 μΜ for 96h (Pani et al.). The calculated IC50 values fall within the same range and the differences observed could be attributed to the different proliferation assays used in each study (CCK-8 and MTT, respectively).

  1. Is resveratrol approved by the U.S. Food and Drug Administration? That information needs to be provided.

Addressed. See end of section 2.1 and below

In the U.S, RSV is sold as a dietary supplement in health food stores and this does not require the Food and Drug Administration (FDA) approval. As per the Federal Food, Drug, and Cosmetic Act (FD&C Act) which was amended in 1994, the FDA does not have the authority to approve any dietary supplements but rather regulate the product after it enters the marketplace. The capsules sold in health food stores contain 200-800 mg RSV and the recommended daily dose of RSV is around 400-800 mg and to not exceed 1000 mg. Hopefully, more clinical studies will be performed in the future to determine the optimal dose of RSV for treating different diseases including cervical cancer.

  1. What are the limitations of using resveratrol as a therapeutic agent?

The limitations of RSV as a therapeutic agent against cancer are its low bioavailability, potential estrogenic activity and lack of clinical studies.

Please see the revised section in 2.1 and  Limitations and controversies.

  1. The estrogenic activity of resveratrol has been reported in a few papers. If that is the case, how the authors can suggest anticancer treatment.

Addressed. See Limitations and controversies section and below

It should be noted that the structure of RSV resembles the structure of the steroid female sex hormone estrogen and for this reason RSV is referred as phytoestrogen. Many studies utilizing breast cancer cells in vitro and animal models of breast cancer have shown RSV to act as an estrogen receptor antagonist and induce anticancer effects [79,80]. However, studies have also reported potential estrogenic activity of RSV. Gehm et al. reported in 1997 that RSV (3-10 µM) inhibited the binding of estradiol to its receptor and increased transcription of estrogen-responsive reporter genes in human breast cancer cells including MCF-7 cells [81]. This effect was estrogen receptor-dependent and was inhibited by specific estrogen antagonists. Overall, this was the first study indicating that RSV acts as estrogen agonist.  In another study, Bowers et al. found that RSV binds to both estrogen receptor alpha (ERα) and beta (ERβ) and acted as an estrogen agonist as seen by the increased ERE-driven reporter gene activity in CHO-K1 cells expressing either receptor [82]. In another study, treatment of breast cancer MCF-7 cells with RSV was found to induce apoptosis by a mechanism involving ERK activation and p53 phosphorylation and acetylation indicating anticancer effects that are p53-dependent [83]. Interestingly, these effects of RSV were blocked in the presence of estrogen [83]. The above studies are performed utilizing in vitro models and unfortunately the estrogenic potential of RSV in animals and humans is unknown. In a pilot study conducted in postmenopausal women with high body mass index (BMI ≥ 25 kg/m2) administration of 1 g of RSV daily for 12 weeks did not result in significant changes in serum concentrations of estradiol, estrone, and testosterone but led to an average of 10% increase in the concentrations of sex steroid hormone binding globulin (SHBG) [84]. This elevation in SHBG could reduce the bioavailability and thus bioactivity of sex steroid hormones including estrogen. Clearly, more animal and clinical studies are required to examine whether RSV acts as estrogen receptor agonist or antagonist in vivo.

  1. What are the adverse effects of resveratrol?

 Addressed. See Limitations and controversies section and below

A few review manuscripts are focused on RSV toxicity and adverse effects in humans [90–92]. Cottart et al. reviewed the existing studies up to 2010 and reported that oral administration of RSV in humans is overall well tolerated. It should be noted however that in the majority of these studies RSV was administered as a single dose of less than 1g in healthy subjects. Administration of RSV at higher doses and/or patients suffering from different conditions may result in different outcomes. Indeed, side effects such as nausea, vomiting, diarrhea and liver dysfunction was seen in patients with non-alcoholic fatty liver disease, at doses of 2.5 g or more per day [93]. Likewise, in a study by Pollack et al. [94] administration of 3 g RSV/daily to older glucose-intolerant adults for 6 weeks resulted  in severe gastrointestinal symptoms that were disappeared when the dose was reduced to 1 g twice daily. Overall, these data indicate that the reported gastrointestinal side effects of RSV are dose dependent. Possible toxicity and adverse effects of long-term (years) RSV administration remains to be examined. 

Reviewer 3 Report

Title: Resveratrol against cervical cancer: evidence from in vitro and in vivo
studies
Authors: Matteo Nadile, Maria Ilektra Retsidou, Katerina Gioti, Apostolos
Beloukas, Evangelia Litsa Tsiani

Summary:

The authors summarize the studies published to date (in vitro/in vivo) on the effect of resveratrol on cervical cancer. The paper is well researched and includes helpful tables and figures. Of particular note, the limitations are also mentioned (e.g., there are no clinical trials yet) in order to develop new research ideas.

Overall, I have the following suggestions for additions:

1. introduction: here some parts should be supported and linked by appropriate references, especially page 1, line 31-37 and page 2, line 60-71.

2. Chapter 1.2: some more information about resveratrol would be interesting for the reader, e.g.: How long has the ingredient been known? How long has it been researched? Are there general vitro/vivo/clinical studies on the listed biological effects?

3. The tables and figures presented are helpful. They should be referenced in the text.

4. Chapter 2.1: Since line 92 mentions a chronological order, please add when the first results were published. The chapter is very long and therefore confusing for the reader. Please check if there is a reasonable way to split the chapter.

5. Chapter 2.2: Please add the reasons why resveratrol analogues are being sought.

6. it is important that the authors address the limitations. To this end, could be added:

(a) line 394-396: 3D models could be used to explore the impact of the tumor microenvironment, as has been done successfully in other tumor types.

Please add an additional reference:

doi: 10.3390/molecules25184292,

doi: 10.3389/fonc.2021.764204.

(b) I miss the mention and discussion of sirtuins throughout the paper. Sirt1 plays a central role in the effect of resveratrol on tumor cells.

Please add an additional reference:

doi: 10.1002/jbt.22625,

doi: 10.3390/nu8030145.

This may also be relevant to cervical cancer - references:

doi: 10.1159/000520642,

doi: 10.1016/j.humpath.2016.09.019.

7.The entire text needs fine-tuning:

(a) References cited by name should all be written as "author et al."

(b) Decisions should be made on which abbreviations to use, and these should be implemented consistently, questionable e.g. at Caspase-3 (lines 116 and 148 ff.).

c) References 53 and 61 are italicized in the text. Please check if they are linked correctly and adjust them if necessary.

d) The chapter numbering needs to be adjusted, currently 4 follows 2.3.

Author Response

Reviewer 3

Summary: The authors summarize the studies published to date (in vitro/in vivo) on the effect of resveratrol on cervical cancer. The paper is well researched and includes helpful tables and figures. Of particular note, the limitations are also mentioned (e.g., there are no clinical trials yet) in order to develop new research ideas.

Overall, I have the following suggestions for additions:

  1. introduction: here some parts should be supported and linked by appropriate references, especially page 1, line 31-37 and page 2, line 60-71.

Thank you for this comment. Addressed. Please see the revised manuscript with the edits.

  1. Chapter 1.2: some more information about resveratrol would be interesting for the reader, e.g.: How long has the ingredient been known? How long has it been researched? Are there general vitro/vivo/clinical studies on the listed biological effects?

The following information has been added in the revised manuscript.

Resveratrol was first isolated in 1939 by Takaoka et al. from Veratrum grandiflorum O (ref) and has extensively been investigated with over 20,000 research papers (including in vitro and in vivo studies) published today.

The first study to examine the anticancer properties of RSV was published  in 1997 by John Pezzuto’s group.

  1. The tables and figures presented are helpful. They should be referenced in the text.

All tables and figures are referenced in the text of the revised manuscript.

  1. Chapter 2.1: Since line 92 mentions a chronological order, please add when the first results were published. The chapter is very long and therefore confusing for the reader. Please check if there is a reasonable way to split the chapter.

Studies that have examined the effects of RSV on different cervical cancer cells in vitro are presented below in chronological order and in Table 1. The first study was published in 2002.

  1. Chapter 2.2: Please add the reasons why resveratrol analogues are being sought.

RSV is a molecule with low bioavailability (discussed in the end of section 2.3) and new analogs are being developed with the goal to increase bioavailability as well as effectiveness while decreasing potential toxicity to healthy/normal tissues.

  1. It is important that the authors address the limitations. To this end, could be added:

  1. line 394-396: 3D models could be used to explore the impact of the tumor microenvironment, as has been done successfully in other tumor types.

Addressed. See Limitations and controversies section and below

Another limitation of the in vitro studies is the lack of use of three-dimensional (3-D) cervical cancer models [75]. Tumor cells in vivo exist in a tumor microenvironment (TME) composed of extracellular matrix (ECM), supporting stromal cells, blood vasculature, and infiltrating immune cells. The interaction between tumor cells and TME plays a pivotal role in tumor development and progression. The use of 3D in vitro tumor models [75] mimics the in vivo TME better and therefore will provide significant data of the potential inhibitory effects of RSV against cervical cancer. 

  1. I miss the mention and discussion of sirtuins throughout the paper. Sirt1 plays a central role in the effect of resveratrol on tumor cells.

Please add an additional reference:

doi: 10.1002/jbt.22625,

doi: 10.3390/nu8030145

This may also be relevant to cervical cancer - references:

doi: 10.1159/000520642,

doi: 10.1016/j.humpath.2016.09.019.

Addressed. See Limitations and controversies section and below

RSV is established as an activator of silent information regulator-1 (SIRT1), a class III histone protein deacetylase (HDACs) involved in many cellular processes [85].  Treatment of colorectal cancer cells with RSV increased SIR1 activation, decreased proliferation, invasion/metastasis and increased in mesenchymal to epithelial transition (MET) [86]. Similarly, RSV activated SIRT1, reduced NFkB and acted against alcohol-aflatoxin B1-induced Hepatocellular carcinoma (HCC)[87].

However, the role of SIRT-1 in cervical cancer is not clear.  Wang et al. found that expression of SIRT1 (measured by Immunohistochemistry) in cervical tumor tissues was increased with cancer progression and a positive correlation existed between SIRT1 and HPV 16 E7 protein levels. In in vitro studies silencing of HPV16 E7 reduced SIRT1 levels. SIRT1 depletion inhibited proliferation, migration and invasion and induced apoptosis of SiHa cervical cancer cells [88]. In a study by Velez -Perez et al., SIRT1 expression was also correlated with progression of cervical squamous cell carcinoma (SCC) [89]. Although these studies [88,89], suggest that SIRT1 may serve as a biomarker for predicting cervical cancer progression, it is not clear if the increased SIRT1 levels are part of compensatory mechanism attempting to counteract cancer progression.

  1. The entire text needs fine-tuning:
    1. References cited by name should all be written as "author et al."
    2. Decisions should be made on which abbreviations to use, and these should be implemented consistently, questionable e.g. at Caspase-3 (lines 116 and 148 ff.).
    3. References 53 and 61 are italicized in the text. Please check if they are linked correctly and adjust them if necessary.
    4. The chapter numbering needs to be adjusted, currently 4 follows 2.3.

Thank you for these comments, they have been all addressed in the manuscript.